# Subclinical articulatory changes of vowel parameters in Korean amyotrophic lateral sclerosis patients with perceptually normal voices

Jin-Ah Kim[1,2,3☉], Hayeun Jang[4☉], Yoonji Choi[5], Young Gi Min[1,2], Yoon-Ho Hong[6], Jung-Joon Sung[1,7]*, Seok-Jin Choi [1,8]*

1 Department of Neurology, Seoul National University Hospital, Seoul, Republic of Korea, 2 Department of Translational Medicine, Seoul National University College of Medicine, Seoul, Republic of Korea, 3 Genomic Medicine Institute, Medical Research Center, Seoul National University, Seoul, Republic of Korea, 4 Division of English, Busan University of Foreign Studies, Busan, Republic of Korea, 5 Department of Korean Language and Literature, Seoul National University, Seoul, Republic of Korea, 6 Department of Neurology, Seoul Metropolitan Government-Seoul National University Boramae Medical Center, Seoul, Republic of Korea, 7 Neuroscience Research Institute, Seoul National University College of Medicine, Seoul, Republic of Korea, 8 Center for Hospital Medicine, Seoul National University Hospital, Seoul, Republic of Korea

☉ These authors contributed equally to this work.
* jjsaint@snu.ac.kr (JJS); seokjin83@gmail.com (SJC)

**Data Availability Statement:** All relevant data are within the paper and its Supporting Information files.

## Abstract

The available quantitative methods for evaluating bulbar dysfunction in patients with amyotrophic lateral sclerosis (ALS) are limited. We aimed to characterize vowel properties in Korean ALS patients, investigate associations between vowel parameters and clinical features of ALS, and analyze subclinical articulatory changes of vowel parameters in those with perceptually normal voices. Forty-three patients with ALS (27 with dysarthria and 16 without dysarthria) and 20 healthy controls were prospectively collected in the study. Dysarthria was assessed using the ALS Functional Rating Scale-Revised (ALSFRS-R) speech subscores, with any loss of 4 points indicating the presence of dysarthria. The structured speech samples were recorded and analyzed using *Praat* software. For three corner vowels (/a/, /i/, and /u/), data on the vowel duration, fundamental frequency, frequencies of the first two formants (F1 and F2), harmonics-to-noise ratio, vowel space area (VSA), and vowel articulation index (VAI) were extracted from the speech samples. Corner vowel durations were significantly longer in ALS patients with dysarthria than in healthy controls. The F1 frequency of /a/, F2 frequencies of /i/ and /u/, the VSA, and the VAI showed significant differences between ALS patients with dysarthria and healthy controls. The area under the curve (AUC) was 0.912. The F1 frequency of /a/ and the VSA were the major determinants for differentiating ALS patients who had not yet developed apparent dysarthria from healthy controls (AUC 0.887). In linear regression analyses, as the ALSFRS-R speech subscore decreased, both the VSA and VAI were reduced. In contrast, vowel durations were found to be rather prolonged. The analyses of vowel parameters provided a useful metric correlated with disease severity for detecting subclinical bulbar dysfunction in ALS patients.

**Funding:** This study was supported by the SNUH Research Fund (0420222070). This work was supported by the National Research Foundation of Korea (NRF) Grant funded by the Korean Government (NRF-2018R1A5A2025964). This study was supported by a grant of the MD-Phd/ Medical Scientist Training Program through the Korea Health Industry Development Institute (KHIDI) funded by the Ministry of Health & Welfare, Republic of Korea. The funders had no role in study design, data collection and analysis, decision to publish, or preparation of the manuscript.

**Competing interests:** The authors have declared that no competing interests exist.

## Introduction

Amyotrophic lateral sclerosis (ALS) is a devastating neurodegenerative disorder characterized by degeneration of upper (UMNs) and lower motor neurons (LMNs) in the brain, brainstem, and spinal cord [1]. Bulbar dysfunction develops in around 30% of people with ALS as their initial symptom [2], and ultimately appears in most cases. The early detection of bulbar impairment is crucial for differentiating ALS mimics, predicting the prognosis, communicating with caregivers, and managing nutritional problems; however, there have been no objective tools for detecting bulbar dysfunction in patients with ALS [3]. The only measures routinely employed in clinical practice are the physician's auditory-perceptual evaluation or the patient's self-report, neurological examination, and needle electromyography [4]. Furthermore, bulbar assessment is often confusing to physicians due to artifacts caused by poor relaxation of bulbar musculature, particularly in those with mild bulbar impairment.

Speech is produced by coordinated actions of articulatory, resonatory, phonatory, and respiratory subsystems [5]. Dysfunction of one or more of these speech subsystems causes several forms of dysarthria in patients with ALS, typically characterized by mixed spastic-flaccid subtype presenting with articulatory imprecision, hypernasality, harshness, slow speaking rate, and prosodic abnormalities, resulting in significant disturbances in communication [5]. Previous studies have repeatedly shown that phonatory measures such as jitter, shimmer, and harmonics-to-noise ratio discriminate ALS patients with bulbar impairment from healthy controls (HCs) [2,4]. However, the articulatory component of speech, rather than the voice quality, impacts communication and thus should be addressed. Clear pronunciation and the acoustic quality of each phoneme, the smallest sound unit that makes language-specific meaning distinctions, are tightly linked to articulation [6]. Speech intelligibility significantly deteriorates when phonemes are not articulated correctly, and the intended meaning is more likely to be misunderstood [7].

A syllable is a basic unit of speech that is made up of vowels, consonants, or a combination of both [8]. Vowels are one of the most important phonemes in speech intelligibility [9]. Since vowels have a longer duration and higher resonance than consonants, they can serve as the nucleus of syllables [10]. The tongue is a primary articulator of vowels, and bulbar muscle weakness in patients with ALS is usually more prominent in the tongue than other bulbar muscles such as the lips and jaw [11,12]. Impairment of vowel articulation has been identified as a potential surrogate marker for neurodegenerative disorders, such as Parkinson's disease (PD), and can even be observed in individuals with rapid eye movement sleep behavior disorder, which is considered a prodromal stage of PD [13,14]. This impairment may worsen as the disease progresses [15]. However, there is a lack of studies specifically investigating vowel articulatory changes in patients with ALS.

In this study, we aimed to characterize vowel properties observed in Korean patients with ALS, investigate the association between vowel parameters and clinical features of ALS, and analyze subclinical articulatory changes of vowel parameters in ALS patients with perceptually normal voices.

## Methods

### Study participants

We prospectively collected speech samples from patients with ALS/motor neuron disease and HCs at Seoul National University Hospital between December 2019 and October 2020. The diagnosis of ALS/motor neuron disease included definite, probable, or possible ALS according to the revised El Escorial criteria [16] and also progressive muscular atrophy, which is

characterized by the degeneration of LMNs without obvious signs of UMN loss [17]. ALS Functional Rating Scale-Revised (ALSFRS-R) were assessed in all 43 patients, and the presence or absence of dysarthria was determined by the speech subscore (4 points) of ALSFRS-R [18]. We investigated the onset region, disease duration, UMN and LMN signs in the bulbar segment, frontal release sign, and forced vital capacity (FVC, % of predicted).

### Standard protocol approvals, registrations, and patient consents

This study was approved by the Institutional Review Board of Seoul National University Hospital (IRB No. 1911-111-108). Informed consent was obtained from all study participants before any procedures. The records of personal information that could identify research subjects will be kept confidential during and after data collection.

### Speech collection

We collected speech samples using a voice recorder (ICD-PX370 Mono Digital Voice Recorder, Sony Corporation, Tokyo, Japan) in a quiet room. The sampling frequency was 44.1 kHz, and the quantization level was set to 16 bits. Experimental participants read the stimuli written in Korean at a natural speed. The stimuli consisted of two types: (i) six repetitions of consonant-vowel (CV) open syllables and (ii) consonant-vowel-consonant (CVC) closed monosyllabic words. A labial lenis stop /p/ was employed as a beginning consonant in a series of CV open syllables repeated six times. Because /p/ is pronounced using both lips as the main articulators, it has little effect on the movement of the tongue body, which is primarily used when pronouncing vowels. Participants were instructed to pronounce each syllable /pi, pu, pa/ six times in a row. In the set of CVC closed monosyllabic words, the corner vowels /i, u, a/ were accompanied by beginning consonants with multiple places of articulation (e.g., labial /p/, alveolar /t, s/, palatal /tɕ/, and velar /k/) and manners (e.g., stop /p, t, k/, fricative /s/, affricate /tɕ/). Following the high corner vowels /i, u/, the closing consonant was unified as a labial nasal /m/. To avoid the monotony of the experimental stimuli, the low back vowel /a/ was followed by several closing consonants: alveolar and velar stops, as well as a liquid /l/. Furthermore, with the liquid closing consonant /l/, /a/ had stop onsets with the laryngeal contrast (for example, lenis /p, t, k, s, t/, tensed /p', t', k', s', tɕ'/, and aspirated /pʰ, tʰ, kʰ, tɕʰ /). S1 Table contains a list of the stimuli words.

### Speech analysis

Each syllable was automatically detected using the *Praat* software, and vowels were marked manually. We measured vowel duration (ms), fundamental frequency (F0, Hz), frequencies of the first two formants (F1 and F2, Hz), harmonics-to-noise ratio (dB), and calculated the vowel space area (VSA, $Hz^2$) and vowel articulation index (VAI, conventional unit). We analyzed the F1 and F2 frequencies of the three vowels (/i/, /u/, and /a/), named corner vowels because they have extreme F1 and F2 values at both end positions of the tongue (Fig 1).

A Korean vowel diagram is shown in Fig 1. Each vowel has unique formants produced by the acoustic resonance of the vocal tracts. The frequencies of the F1 and F2 have been regarded as being the most relevant to the production and perception of vowels [19]. The frequency of F1 is determined by the tongue's position on the vertical axis; the frequency of F2 is determined by the tongue's position on the horizontal axis and the length of the oral cavity. VSA refers to the two-dimensional area formed by lines connecting F1 and F2 frequencies of corner vowels [20]. In patients with an articulatory problem, the value of VSA is usually reduced with the decreasing high formant frequencies and/or with the increasing low formant frequencies, called vowel formant centralization [21]. The VAI is designed to be as insensitive to

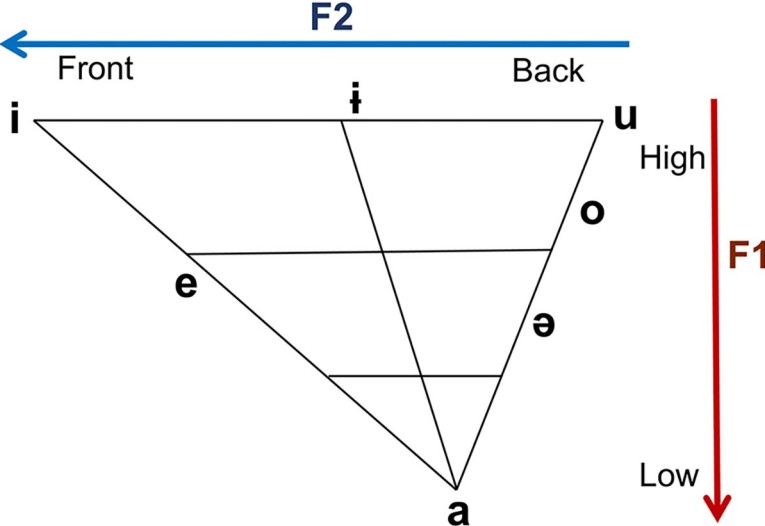

**Fig 1. Korean vowel diagram.** The tongue moves vertically and horizontally within the oral cavity, and corner vowels (/a/, /i/, and /u/) differ in the position of the tongue when pronounced. Each vowel in the vowel diagram has a unique first (F1) and second formant (F2).

interspeaker variability as possible while being as sensitive as possible to vowel formant centralization. Therefore, we additionally calculated each speaker's VAI as (F2 of /i/ + F1 of /a/) / (F2 of /u/ + F2 of /a/ + F1 of /u/ + F1 of /i/) [22].

## Statistical analysis

For comparison of clinical features, we performed the Mann-Whitney U test or Kruskal-Wallis H test for continuous variables and the Pearson's chi-square test for categorical variables. We compared group differences in vowel parameters using a one-way analysis of covariance (ANCOVA) with Tukey's HSD post-hoc tests with age and sex adjustment. Then we investigated associations of vowel parameters with clinical variables of ALS/motor neuron disease by multivariable linear regression analyses. In each model, we set vowel parameters with $p < 0.05$ in ANCOVA tests (vowel duration, F1/F2 frequencies, VSA, and VAI) as outcome variables. We tested the effects of explanatory variables (FVC and speech subscore of ALSFRS-R) after adjusting for age and sex. UMN and LMN signs were not included in the explanatory variables to avoid the multicollinearity problem. The diagnostic performance of selected vowel parameters was assessed by receiver operating characteristic (ROC) curve analysis using logistic regression with a generalized linear model. A two-tailed $p < 0.05$ was considered statistically significant. All statistical analyses were performed using the R software version 4.1.0 (The R Foundation for Statistical Computing Platform) [23,24].

## Results

### Clinical characteristics

A total of 43 patients with ALS/motor neuron disease [27 with dysarthria (ALSwD) and 16 without dysarthria (ALSwoD)] and 20 HCs were enrolled. Table 1 summarizes the demographic information and clinical characteristics of study participants. The female to male ratio was significantly lower in the ALSwoD group than in the other two groups (ALSwD 55.6%, ALSwoD 12.5%, HC 50.0%, $p$ = .016). The median age of HCs seemed younger than ALS patients with or without dysarthria, but there were no statistically significant differences

**Table 1. Demographic and clinical characteristics of the study participants.**

| Characteristic | ALSwD (n = 27) | ALSwoD (n = 16) | Control (n = 20) | P value |
|---|---|---|---|---|
| Sex, female (%) | 15 (55.6) | 2 (12.5) | 10 (50) | **.016** |
| Age (years)* | 60.0 [50.0;66.0] | 62.5 [48.5;67.0] | 54.5 [42.0;60.5] | .077 |
| Disease duration (years)* | 2.0 [1.0;3.5] | 2.0 [1.0;5.0] | | .768 |
| Onset region | | | | **.003** |
| Bulbar (%) | 12 (44.4) | 0 (0.0) | | |
| Cervical (%) | 8 (29.6) | 12 (75) | | |
| Lumbosacral (%) | 7 (25.9) | 4 (25) | | |
| ALSFRS-R* | | | | |
| Total | 37.5 [33.0;42.0] | 38.5 [35.0;41.0] | | .736 |
| Bulbar subscore | 9.0 [8.0;10.0] | 12.0 [11.0;12.0] | | < **.001** |
| Speech subscore | 3.0 [2.0;3.0] | 4.0 [4.0;4.0] | | < **.001** |
| FVC (% predicted)* | 78.0 [70.0;88.0] | 82.0 [68.0;90.0] | | .836 |
| UMN sign, bulbar (%) | 23 (85.2) | 6 (37.5) | | **.004** |
| LMN sign, bulbar (%) | 21 (77.8) | 1 (6.2) | | < **.001** |
| Frontal release sign (%) | 11 (40.7) | 1 (6.2) | | **.037** |
| Diagnosis | | | | **.008** |
| Definite ALS | 11 (40.7) | 1 (6.2) | | |
| Probable ALS | 11 (40.7) | 6 (37.5) | | |
| Possible ALS | 5 (18.5) | 5 (31.2) | | |
| PMA | 0 (0.0) | 4 (25.0) | | |

Data are expressed as the number of subjects (%). *Data are expressed as median [25th and 75th percentiles].

Significant findings with P<.050 are in **bold** fonts.

Abbreviations: ALS, amyotrophic lateral sclerosis; ALSwD, ALS with dysarthria; ALSwoD, ALS without dysarthria; ALSFRS-R, ALS Functional Rating Scale-Revised; FVC, forced vital capacity; UMN, upper motor neuron; LMN, lower motor neuron; PMA, progressive muscular atrophy.

(ALSwD 60.0, ALSwoD 62.5, HC 54.5 years old, $p = .077$). The predicted values of FVC were slightly reduced both in the ALSwD and ALSwoD groups compared to the reference values (cut-off 80%), but there was no group difference. The total ALSFRS-R score was similar between the ALSwD and ALSwoD groups. By group definition, the speech subscore of ALSFRS-R was 4.0 in the ALSwoD group, which was significantly higher than 3.0 in the ALSwD group ($p < .001$). Both the bulbar UMN and LMN signs were significantly more frequent in the ALSwD compared to the ALSwoD group. The proportion of definite ALS was higher in the ALSwD group, whereas possible ALS and PMA were more common in the ALSwoD group.

## Comparison of vowel parameters between groups

Table 2 shows vowel parameters of three corner vowels, /a/, /i/, and /u/ in each group. Vowel duration of all three vowels were markedly prolonged in the ALSwD group than those in ALSwoD and HCs ($p < .001$ for all comparisons). The F1 frequencies of /a/ in the ALSwD and ALSwoD groups were significantly lower compared to HCs (ALSwD vs. HC, $p < .001$; ALSwoD vs. HC, $p < .001$). The F2 frequencies of /i/ in the ALSwD group were notably lower than those in HCs ($p < .012$). Those of /u/ in the ALSwD group were markedly longer than in the ALSwoD and HCs (ALSwD vs. ALSwoD, $p < .048$; ALSwD vs. HC, $p < .021$). The VSA was significantly reduced in ALSwD and ALSwoD compared to HCs (ALSwD vs. HC, $p < .001$; ALSwoD vs. HC, $p < .014$). The VAI was significantly lower in ALSwD compared to HCs, and it also showed a slight reduction, though not statistically significant, in ALSwoD compared to

**Table 2. Comparisons of vowel parameters between groups.**

| Parameter | ALSwD (n = 27) | ALSwoD (n = 16) | Control (n = 20) | P value | | | |
|---|---|---|---|---|---|---|---|
| | | | | Overall | ALSwD vs. ALSwoD | ALSwD vs. Control | ALSwoD vs. Control |
| Duration (ms) | | | | | | | |
| /a/ | 129.9 (32.6) | 81.2 (13.8) | 81.0 (15.7) | **< .001** | **< .001** | **< .001** | NS |
| /i/ | 171.7 (49.2) | 98.0 (20.7) | 118.0 (35.6) | **< .001** | **< .001** | **< .001** | NS |
| /u/ | 158.2 (46.1) | 91.5 (21.1) | 99.0 (28.9) | **< .001** | **< .001** | **< .001** | NS |
| F0 (Hz) | | | | | | | |
| /a/ | 157.55 (36.6) | 159.3 (28.0) | 157.58 (42.4) | .98 | NS | NS | NS |
| /i/ | 157.9 (37.0) | 166.6 (28.5) | 161.6 (45.4) | .47 | NS | NS | NS |
| /u/ | 161.0 (39.4) | 165.1 (28.6) | 162.3 (43.5) | .88 | NS | NS | NS |
| F1 (Hz) | | | | | | | |
| /a/ | 764.4 (94.2) | 741.9 (69.2) | 818.6 (96.4) | **< .001** | NS | **< .001** | **< .001** |
| /i/ | 382.6 (57.5) | 405.2 (70.6) | 397.3 (72.7) | .47 | NS | NS | NS |
| /u/ | 458.1 (82.1) | 437.7 (65.5) | 446.8 (55.2) | .68 | NS | NS | NS |
| F2 (Hz) | | | | | | | |
| /a/ | 1418.9 (161.4) | 1389.3 (128.8) | 1445.2 (197.9) | .42 | NS | NS | NS |
| /i/ | 2038.3 (258.9) | 2072.1 (183.6) | 2207.2 (290.7) | **.02** | NS | **.012** | NS |
| /u/ | 1329.5 (248.6) | 1155.9 (185.8) | 1152.1 (173.7) | **.01** | **.048** | **.021** | NS |
| Harmonics-to-noise ratio (dB) | | | | | | | |
| /a/ | 7.8 (5.3) | 8.4 (5.2) | 10.3 (2.9) | .23 | NS | NS | NS |
| /i/ | 6.3 (18.5) | 12.6 (3.2) | 11.4 (18.4) | .42 | NS | NS | NS |
| /u/ | 8.7 (21.8) | 11.0 (6.9) | 15.0 (7.2) | .42 | NS | NS | NS |
| Vowel space area (Hz$^2$) | 128183.9 (77881) | 144782.7 (66485) | 210574.1 (90416) | **< .001** | NS | **< .001** | **.014** |
| Vowel articulation index (conventional unit) | 0.787 | 0.831 | 0.883 | **.003** | NS | **.002** | **NS** |

Data are expressed as mean (standard deviation).

Significant findings with *P*<.050 are in **bold** fonts.

Abbreviations: ALS, amyotrophic lateral sclerosis; ALSwD, ALS with dysarthria; ALSwoD, ALS without dysarthria; NS, not significant.

HCs. There were no discernible group differences observed in fundamental frequency and harmonics-to-noise ratio.

## Association between vowel parameters and clinical variables

The linear regression analyses showed that, as the speech subscore of ALSFRS-R decreased, durations of all three corner vowels were correspondingly prolonged (*p* < .001 for /a/; *p* = .022 for /i/; *p* = .012 for /u/) (Fig 2A), whereas the VSA decreased along with the ALSFRS-R speech subscore in both sexes (*p* = .038 for male; *p* = .007 for female) (Fig 2B). The VAI decreased along with the ALSFRS-R speech subscore, as did the VSA (S1 Fig). The detailed data on the associations between vowel parameters and clinical variables are described in Table 3. Corner vowel durations were significantly longer in patients who had both the UMN and LMN signs than in those who did not (S2 Table). In addition, durations of corner vowels appeared more severely impacted by the LMN signs than UMN signs, although there were no statistically significant differences (Fig 3).

## Diagnostic performance of vowel parameters

In the ROC curve analyses for ALSwD and HCs, the area under the curve (AUC) for the duration of /a/, /i/, and /u/ was 0.733, 0.822, and 0.800, respectively (S3 Table and S2A Fig). The

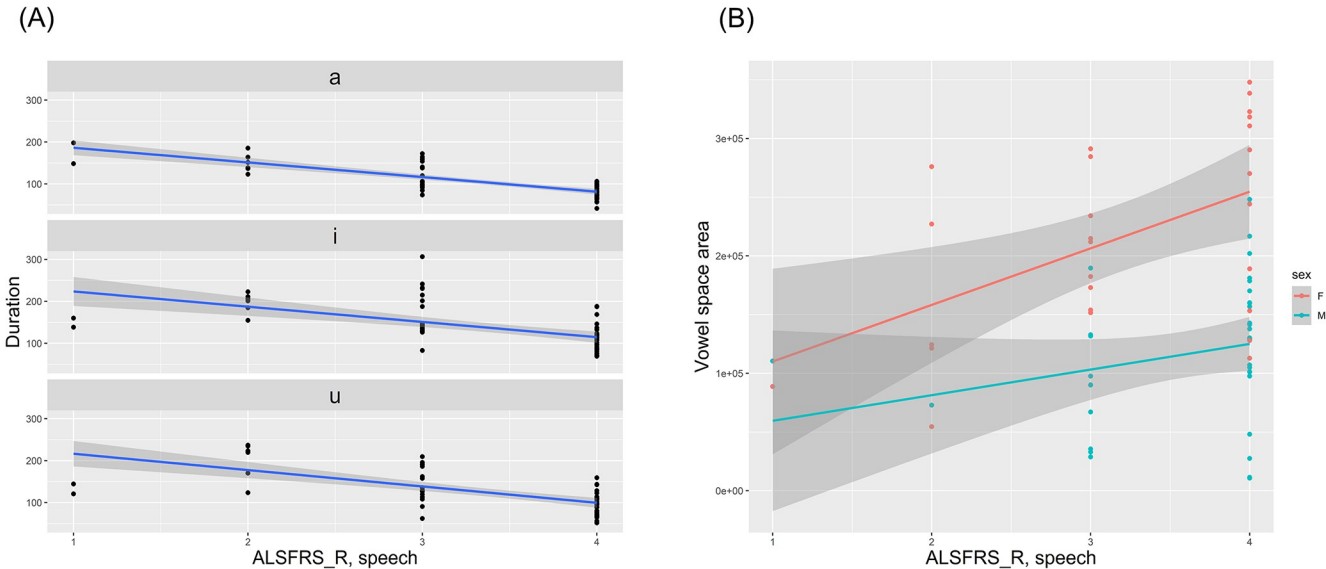

**Fig 2. Association between ALSFRS-R speech subscore and vowel parameters.** As the ALSFRS-R subscore decreased, the durations of three corner vowels were inversely prolonged (A). The VSA decreased along with the ALSFRS-R subscore both in male and female (B). Abbreviations: ALS, amyotrophic lateral sclerosis; ALSFRS-R, ALS Functional Rating Scale-Revised; VSA, vowel space area.

AUC for F1 frequency of /a/, F2 frequencies of /i/ and /u/, and VSA was 0.557, 0.538, 0.693, and 0.643, respectively (S3 Table and S2B Fig). Taking all these variables together, we obtained an improved diagnostic value with an AUC of 0.912 (Fig 4A). We then examined the ROC curves in ALSwoD and HCs to determine the diagnostic values of F1 frequency of /a/ and VSA for detecting the subclinical articulatory change in ALS patients with perceptually normal voices. As a result, the AUC for F1 frequency of /a/ and VSA were 0.887 and 0.809, respectively (Fig 4B and 4C).

## Discussion

In this study, we attempted to analyze corner vowel parameters in patients with ALS/motor neuron disease and HCs, find out the associations between vowel parameters and clinical features of ALS, and identify subclinical changes in vowel parameters in those with perceptually normal voices. We showed that the F1 frequencies of /a/ and VSA were the most reliable parameters for diagnosing ALS patients without apparent dysarthria but presumably developing subclinical bulbar impairment. Furthermore, the VSA and vowel durations were significantly associated with the disease severity, such as the ALSFRS-R speech subscore. Our findings provide preliminary data on vowel analysis as a bulbar measure for patients with ALS.

In ALS patients with dysarthria, durations of all corner vowels were considerably longer than those in HCs. Although lips and teeth play a role in the production of vowels, the tongue is the primary articulator of vowels [11]. Thus, prolonged vowel durations in patients with ALS can be interpreted as a sign of impaired tongue movement [25]. On the other hand, given the similar vowel durations between ALS patients without dysarthria and HCs, vowel durations do not seem to be an early indicator of bulbar dysfunction. In linguistics studies, the duration can be measured based on the unit of sentence, syllable, or vowel [4]. Previous studies showed that articulatory rates measured by syllables per second are slower, and phrase durations are longer in patients with ALS than those in HCs [26,27]. Conversely, a recent study showed that, in natural speech, mean speech segment durations between pauses are

**Table 3. Results of linear regression analyses.**

| Variable | Estimate | Standard error | T value | P value |
|---|---|---|---|---|
| Duration of /a/ ~ | | | | |
| (Intercept) | 221.465 | 35.342 | 6.266 | |
| Age | -0.176 | 0.357 | -0.492 | .627 |
| Sex (male) | -6.219 | 10.655 | -0.584 | .565 |
| FVC | -0.035 | 0.287 | -0.123 | .903 |
| ALSFRS-R, speech | -30.061 | 5.736 | -5.241 | < **.001** |
| Duration of /i/ ~ | | | | |
| (Intercept) | 220.389 | 72.139 | 3.055 | |
| Age | -0.084 | 0.729 | -0.116 | .909 |
| Sex (male) | -21.376 | 21.748 | -0.983 | .335 |
| FVC | 0.352 | 0.587 | 0.599 | .555 |
| ALSFRS-R, speech | -28.703 | 11.707 | -2.452 | **.022** |
| Duration of /u/ ~ | | | | |
| (Intercept) | 227.760 | 64.906 | 3.509 | |
| Age | -0.023 | 0.656 | -0.035 | .972 |
| Sex (male) | -26.129 | 19.567 | -1.335 | .194 |
| FVC | 0.008 | 0.528 | 0.016 | .988 |
| ALSFRS-R, speech | -27.403 | 10.533 | -2.602 | **.012** |
| F1 of /a/ ~ | | | | |
| (Intercept) | 893.020 | 82.598 | 10.812 | |
| Age | 0.007 | 0.834 | 0.009 | .993 |
| Sex (male) | -132.861 | 24.901 | -5.336 | < **.001** |
| FVC | -0.104 | 0.672 | -0.155 | .878 |
| ALSFRS-R, speech | -16.538 | 13.405 | -1.234 | .229 |
| F2 of /i/ ~ | | | | |
| (Intercept) | 1843.699 | 280.062 | 6.583 | |
| Age | 3.249 | 2.829 | 1.149 | .262 |
| Sex (male) | -342.735 | 84.431 | -4.059 | < **.001** |
| FVC | -0.870 | 2.278 | -0.382 | .706 |
| ALSFRS-R, speech | 90.382 | 45.451 | 1.989 | .058 |
| F2 of /u/ ~ | | | | |
| (Intercept) | 1900.834 | 352.420 | 5.394 | |
| Age | -3.946 | 3.559 | -1.109 | .279 |
| Sex (male) | 77.492 | 106.244 | 0.729 | .473 |
| FVC | -2.053 | 2.866 | -0.716 | .481 |
| ALSFRS-R, speech | -95.802 | 57.193 | -1.675 | .107 |
| VSA ~ | | | | |
| (Intercept) | -37438.0 | 101562.2 | -0.369 | |
| Age | 1435.7 | 817.3 | 1.757 | .088 |
| Sex (male) | -92781.1 | 20769.5 | -4.467 | < **.001** |
| FVC | 317.5 | 747.9 | 0.425 | .675 |
| ALSFRS-R, speech | 39129.4 | 19019.6 | 2.057 | **.047** |
| VAI ~ | | | | |
| (Intercept) | 0.579 | 0.123 | 4.694 | |
| Age | 0.002 | 0.001 | 1.565 | .130 |
| Sex (male) | -0.076 | 0.037 | -2.051 | .051 |
| FVC | <0.001 | 0.001 | -0.012 | .990 |

(*Continued*)

**Table 3.** (Continued)

| Variable | Estimate | Standard error | *T* value | *P* value |
|---|---|---|---|---|
| ALSFRS-R, speech | 0.051 | 0.020 | 2.551 | **.018** |

Significant findings with *P*<.050 are in **bold** fonts.

Abbreviations: ALS, amyotrophic lateral sclerosis; ALSFRS-R, ALS Functional Rating Scale-Revised; FVC, forced vital capacity.

significantly reduced in patients with ALS-frontotemporal dementia compared to HCs, regardless of their bulbar symptoms; they suggested that speech duration is associated with the severity of cognitive impairment [28]. We herein used structured speech stimuli to minimize the influence of cognitive dysfunction, thereby focusing on articulatory changes in vowel parameters.

The F1 frequency of /a/ and the F2 frequencies of /i/ and /u/ were significantly different between ALS patients with dysarthria and HCs. The F1 frequency increases as the tongue moves downward, whereas the F2 frequency reflects the tongue's back-and-forth movement [29]. The above three formants (F1 frequency of /a/ and F2 frequencies of /i/ and /u/) correspond to the vertex in the vowel diagram (Fig 1). These findings show that changes in the formant frequencies in ALS patients with dysarthria are maximized at the vowels having the extreme values of formant frequencies. Furthermore, the decreased F1 frequency of /a/, decreased F2 frequency of /i/ and increased F2 frequency of /u/ all contributed to a reduction in the VSA, which captures articulatory working space. The VSA has been suggested as a

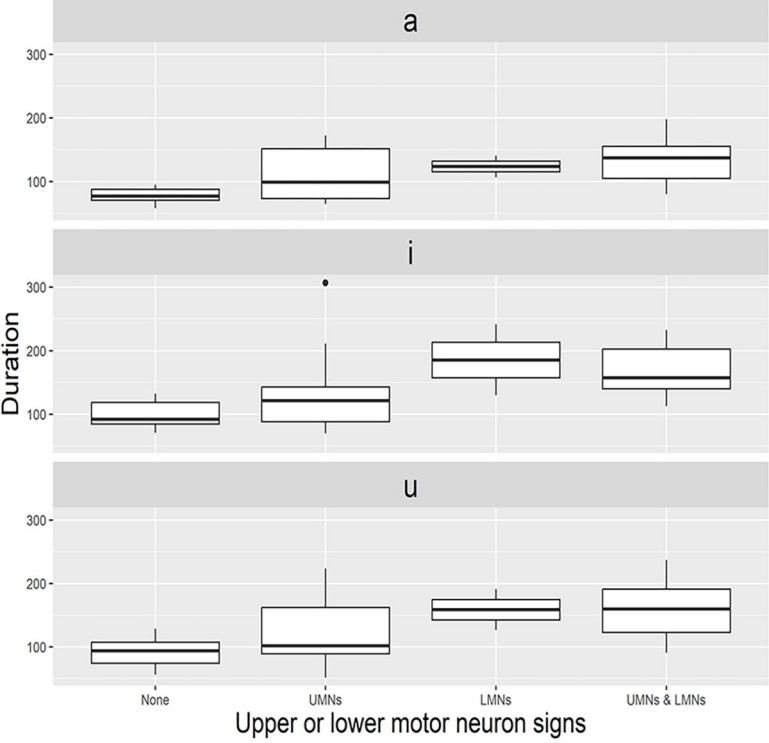

**Fig 3. Changes of corner vowel durations according to the bulbar UMN or LMN signs.** The durations of three corner vowels were significantly longer in patients who had both the UMN and LMN signs than in those who did not. Patients with solely LMN signs had longer vowel durations than those with UMN signs, but there were no statistically significant differences. Abbreviations: UMN, upper motor neuron; LMN, lower motor neuron.

(A)                                    (B)                                    (C)

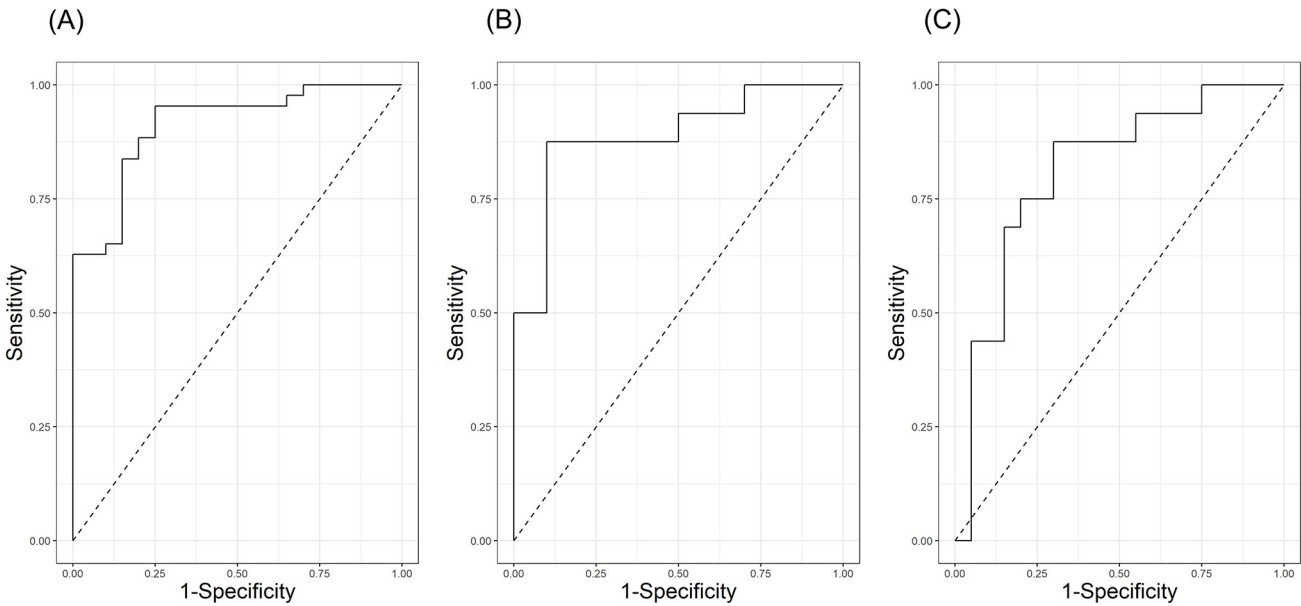

**Fig 4. Receiver operating characteristic curves.** The AUC for three vowel durations, F1 frequency of /a/, F2 frequencies of /i/ and /u/, and VSA between ALSwD and HC was 0.912, with a sensitivity of 95.3% and specificity of 75.0% (A). The AUCs for F1 frequency of /a/ and VSA between ALSwoD and HC were 0.887 (sensitivity 87.5% and specificity 90.0%) and 0.809 (sensitivity 87.5% and specificity 70.0%), respectively (B and C). Abbreviations: AUC, area under the curve; VSA, vowel space area; ALS, amyotrophic lateral sclerosis; ALSwD, ALS with dysarthria; HC, healthy control; ALSwoD, ALS without dysarthria.

quantitative index of speech intelligibility, the degree to which speech sounds can be correctly identified and understood by listeners [30]. In our analysis, the VSA was significantly associated with the severity of dysarthria, which is represented by the ALSFRS-R speech subscore. Therefore, the reduction in VSA may be a useful metric for diagnosing patients with ALS.

We further analyzed changes in vowel parameters in patients with ALS who have not presented with apparent dysarthria yet. As a result, the F1 frequency of /a/ and VSA were identified as reliable measures for detecting subclinical bulbar dysfunction. The genioglossus, the largest of the four extrinsic and four intrinsic muscles in the tongue, is the main protrusor, whereas the hyoglossus functions as a main depressor and retractor [31]. Thus, the reduction in the F1 frequency of /a/ in patients with ALS reflects the restricted vertical movement of the tongue, which might be caused by denervation changes in the hyoglossus. A dissociated muscle atrophy in ALS would provide a plausible explanation for this finding. It is typically observed in the hands of ALS patients, more severe in the lateral hand group of muscles (the first dorsal interosseous and abductor pollicis brevis), with relative preservation of the medial hand (the abductor digit minimi). This split-hand syndrome is believed to result from the associated degeneration of UMNs and LMNs [32]. In addition, other split syndromes such as split-hand plus, split-elbow, and split-leg phenomenon have been described in ALS involving upper or lower limbs [33–35]. In this regard, a preferential degeneration of the hyoglossus may indicate disproportionate muscle atrophy in the tongue, presumably referred to as the *'split-tongue phenomenon*,' which should be further investigated through electrophysiological studies. Besides, a decrease in the F1 frequency of /a/ may contribute to a decrease in the VSA, and impaired coordination of tongue movements may have an additional effect. While a previous study found early phonatory abnormalities (jitter, shimmer, and signal-to-noise ratio) in ALS patients without dysarthria [36], we focused on articulatory changes in those without apparent dysarthria.

The durations of corner vowels were significantly prolonged in patients who had both the UMN and LMN signs in the bulbar segment. Moreover, vowel durations were slightly longer in patients with only LMN signs than those with UMN signs, although it did not reach statistical significance (Fig 3). These findings suggest that dysfunction of the articulatory speech subsystem associated with limited tongue movement was more heavily influenced by the LMN degeneration rather than UMN degeneration. In addition, there were no group differences in the F0 frequency and harmonics-to-noise ratio. Previous studies have shown conflicting results regarding the F0 frequency associated with voice pitch. A few studies reported that a decrease in F0 frequency in patients with ALS is caused by laryngeal muscle weakness [37,38], whereas another study found that the F0 frequency increases due to the impairment of intrinsic laryngeal muscles [39]. Further, a recent study showed an association between the F0 frequency and the Penn UMN bulbar subscale, suggesting that impairment of laryngeal muscles caused by the UMN degeneration may interfere with F0 frequency [28]. For a harmonics-to-noise ratio, which reflects a turbulent noise caused by incomplete glottic closure during sound production [40], a previous study found an increase of harmonics-to-noise ratio in patients with ALS, which means a phonatory instability [39]. In this study, as we focused on articulatory changes of vowel parameters using structured, repetitive speech stimuli, dysfunctions of resonatory or phonatory subsystems might have been underestimated.

While not statistically significant, we acknowledge that patients with ALS were slightly older than the HCs, potentially conferring an advantage in the analysis. Nevertheless, we anticipate that our findings will remain unaffected by potential age differences for the following reasons: Regarding the formant frequencies of the corner vowels, a recent study revealed a decline in F2 /a/ and F2 /u/ with age, while F1 remained unchanged [41]. However, our study showed that F1 /a/ and F2 /i/ decreased, while F2 /u/ rather increased in older ALS patients compared to HCs, showing contradictory results on F2 /u/. Furthermore, some studies reported that the formant frequencies of the corner vowels change minimally across several decades of adult life, with no significant differences observed between middle-aged women (40 to 60 years) and older women (70 to 92 years) [42,43]. Therefore, the 5.5-year difference between ALSwD and HCs, as well as the 8-year difference between ALSwoD and HCs, is unlikely to impact our results on formant frequencies. Additionally, the VSA is known to show a compensatory increase with age [41]. However, in our study, we observed that VSA rather decreased in relatively older ALS patients compared to HCs, supporting our results of a reduced VSA inherent to the disease. Lastly, vowel durations exhibit a slight increase of no more than 30 ms with age, particularly in individuals aged over 70 years [41]. However, when comparing the significant differences in vowel durations between patients with ALS and HCs in our study—exceeding 50 ms for all corner vowels—it becomes evident that vowel durations are longer in patients with ALS, even after adjusting for age differences.

The limitations of the study should be acknowledged. First, our study is limited by the small number of study participants. Second, the female to male ratio was much lower in the ALSwoD group than in HCs. In general, both the F0 frequency and VSA in females are higher (or larger) than males [44,45], and females have higher formant frequencies than males [46], albeit there are some conflicting results. To overcome this issue, we performed the following analyses: (i) We tried to adjust sex differences by conducting an ANCOVA analysis. (ii) The male subgroup analysis generated the same results (S4 Table). (iii) The VSA was also a significant determinant in differentiating the ALSwD group from HCs with a similar sex ratio. Third, as we recorded participants' voices directly without using a head-mounted microphone, standardizing the vowel intensity in our recording setting became challenging. Consequently, we excluded the vowel intensity parameter from the analysis. In addition, it is important to interpret our results on the harmonics-to-noise ratio with caution due to potential recording quality issues

resulting from noise disturbances caused by the absence of a head-mounted microphone and low voice intensity. Fourth, cognitive function was not tested in the study participants. Although cognitive impairment inevitably affects the results of speech analysis, we used structured repetitive speech stimuli to minimize the influence of cognitive dysfunction on articulatory changes. Fifth, because we did not obtain longitudinal data on speech samples, we could not analyze changes in vowel parameters as the disease progressed. Further follow-up studies with a larger number of patients are warranted.

## Conclusions

In conclusion, the analysis of vowel parameters provided a useful metric correlated with disease severity for detecting subclinical bulbar dysfunction in patients with ALS. Because the vowel systems in most languages can be represented by a vowel diagram, with particular similarities in corner vowels [47], the findings of this study would possibly be extended to other languages warranting further investigations.

## Supporting information

**S1 Fig. Association between ALSFRS-R speech subscore and vowel articulation index.**
(PDF)

**S2 Fig. Receiver operating characteristic curves for differentiating ALS patients with dysarthria from healthy controls using each vowel parameter.**
(PDF)

**S1 Table. List of stimuli words.**
(DOCX)

**S2 Table. Effects of bulbar UMNs or LMNs on vowel parameters.**
(DOCX)

**S3 Table. Diagnostic performance of individual vowel parameters in distinguishing ALS patients with dysarthria from healthy controls.**
(DOCX)

**S4 Table. Comparisons of vowel parameters between groups in male subjects.**
(DOCX)

**S1 File. Raw acoustic data.**
(CSV)

**S2 File. Clinical information and extracted vowel parameters.**
(CSV)

## Author Contributions

**Conceptualization:** Jin-Ah Kim, Hayeun Jang, Yoonji Choi, Seok-Jin Choi.

**Data curation:** Jin-Ah Kim, Hayeun Jang.

**Formal analysis:** Jin-Ah Kim, Hayeun Jang.

**Investigation:** Jin-Ah Kim, Hayeun Jang, Seok-Jin Choi.

**Methodology:** Jin-Ah Kim, Young Gi Min, Seok-Jin Choi.

**Software:** Hayeun Jang.

**Validation:** Jin-Ah Kim, Hayeun Jang, Yoon-Ho Hong, Jung-Joon Sung, Seok-Jin Choi.

**Writing – original draft:** Jin-Ah Kim, Hayeun Jang.

**Writing – review & editing:** Yoonji Choi, Young Gi Min, Yoon-Ho Hong, Jung-Joon Sung, Seok-Jin Choi.

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
