## [Decision Letter · Decision Letter 0]

31 May 2023

PONE-D-23-09560Subclinical articulatory changes of vowel parameters in Korean amyotrophic lateral sclerosis patients with perceptually normal voicesPLOS ONE

Dear Dr. Choi,

Thank you for submitting your manuscript to PLOS ONE. After careful consideration, we feel that it has merit but does not fully meet PLOS ONE’s publication criteria as it currently stands. Therefore, we invite you to submit a revised version of the manuscript that addresses the points raised during the review process.

We look forward to receiving your revised manuscript.

Kind regards,

Kyung-Wan Baek, Ph.D.

Academic Editor

PLOS ONE

Journal Requirements:

Additional Editor Comments:

Please respond to the reviewers' opinions and submit revised manuscripts that reflect the review opinions.

Reviewers' comments:

Reviewer's Responses to Questions

**Comments to the Author**

1. Is the manuscript technically sound, and do the data support the conclusions?

Reviewer #1: Yes

Reviewer #2: Yes

2. Has the statistical analysis been performed appropriately and rigorously? 

Reviewer #1: Yes

Reviewer #2: Yes

3. Have the authors made all data underlying the findings in their manuscript fully available?

Reviewer #1: Yes

Reviewer #2: Yes

4. Is the manuscript presented in an intelligible fashion and written in standard English?

Reviewer #1: Yes

Reviewer #2: Yes

5. Review Comments to the Author

Reviewer #1: General Summary

Thank you for the opportunity to review a manuscript titled "Subclinical articulatory changes of vowel parameters in Korean amyotrophic lateral sclerosis patients with perceptually normal voices." In the study, the authors strive to provide a pilot description of vowel articulation deficits in amyotrophic lateral sclerosis (ALS); for the study, authors recorded consonant-vowel (CV) and consonant-vowel-consonant (CVC) utterances from 43 patients diagnosed with definite, probable, or possible ALS and 20 healthy control (HC) participants. The authors found a significant increase in the length of the corner vowels and alterations of the first two format frequencies, as well as the vowel space area (VSA). The authors also performed a classification experiment reaching the area under the curve (AUC) of 0.912.

Generally, authors provide scientifically sound manuscripts with the most detailed descriptions of their experiments. The study design shows few shortcomings, especially considering the dataset; however, authors strive to deal with the issues when possible and discuss possible impacts of the study's limitations.

Suggestions and questions

1) The authors performed a classification experiment, however, reaching interesting AUC and also performed the linear regression analysis; thus, I would expect that the classification experiment was based on logistic regression, but as it is not clearly stated, it may not be apparent and suggest including methodology part where authors describe the classification scenario a validation method.

2) The major weakness of the study is the age and sex match of HC participants. The authors strived to remove the effect of age and sex in the statistical analysis and included them in the linear regression analysis. Moreover, the authors acknowledge the possible confounding effect of sex differences in the limitations. Even though the age difference did not reach the significance level, it is the border value with p = 0.08, and therefore it would be beneficial to discuss it also. Especially because the HC group is younger and thus be advantaged in the analysis. Authors should discuss the results of their linear regression in that manner.

3) How was the measurement of intensity standardized? If I understand correctly, participants were recorded directly by the device, not by the for-instance head-mounted microphone.

4) What are units of harmonic-to-noise ratio? If it is the Praat value, I would expect decibels, but the values listed in the results are low. It could be because the authors recorded lower sound quality by the device. If this is true, I would suggest the speech recording guideline, which also provides recording setup information. (Rusz, J., Tykalova, T., Ramig, L. O., & Tripoliti, E. (2021). Guidelines for speech recording and acoustic analyses in dysarthrias of movement disorders. Movement Disorders, 36(4), 803-814.)

5) What was the recording setup? What was the sampling frequency and level of quantization?

6) Why authors did not include Vowel Articulation Index?

7) At the end of the page, 14 authors discuss the speech rate; they may find interesting studies

a. Novotny, M., Melechovsky, J., Rozenstoks, K., Tykalova, T., Kryze, P., Kanok, M., ... & Rusz, J. (2020). Comparison of automated acoustic methods for oral diadochokinesis assessment in amyotrophic lateral sclerosis. Journal of Speech, Language, and Hearing Research, 63(10), 3453-3460.

b. Rong, P. (2020). Automated acoustic analysis of oral diadochokinesis to assess bulbar motor involvement in amyotrophic lateral sclerosis. Journal of Speech, Language, and Hearing Research, 63(1), 59-73.

Conclusion

To conclude, the author's scientifically sound, comprehensively written manuscript on an exciting topic. The authors acknowledge that this is a pilot study with several shortcomings. The study still needs improvement before it can be published, and some of the modifications can be timely. Therefore I suggest a major revision of the manuscript before publication.

Reviewer #2: The manuscript aims to characterize vowel properties in Korean ALS patients, investigate associations between vowel parameters and clinical features of ALS, and analyze subclinical articulatory changes

of vowel parameters in those with perceptually normal voices. In general, the article is well written with good scientific novelty. However, some methodological issues need to be clarified, including the unbalanced age of healthy control participants. The motivation of study also needs to be improved.

Abstract:

Please add the information that patients were separated into dysarthria and without dysarthria groups based on ALS Functional Rating Scale-Revised speech item.

Introduction:

The motivation of the study needs to be improved. I would suggest going beyond the ALS studies and improving the motivation based on other motor speech disorders. For example, the possibility of detecting the progression of the disease or subclinical articulatory changes has already been shown in Parkinson's disease patients with no perceptible dysarthria or even in patients with REM sleep behaviour disorder. See studies:

Skrabal D, Rusz J, Novotny M, Sonka K, Ruzicka E, Dusek P, Tykalova T. Articulatory undershoot of vowels in isolated REM sleep behavior disorder and early Parkinson's disease. NPJ Parkinsons Dis. 2022 Oct 20;8(1):137. doi: 10.1038/s41531-022-00407-7. PMID: 36266347; PMCID: PMC9584921.

Rusz J, Cmejla R, Tykalova T, Ruzickova H, Klempir J, Majerova V, Picmausova J, Roth J, Ruzicka E. Imprecise vowel articulation as a potential early marker of Parkinson's disease: effect of speaking task. J Acoust Soc Am. 2013 Sep;134(3):2171-81. doi: 10.1121/1.4816541. PMID: 23967947.

Skodda S, Grönheit W, Schlegel U. Impairment of vowel articulation as a possible marker of disease progression in Parkinson's disease. PLoS One. 2012;7(2):e32132. doi: 10.1371/journal.pone.0032132. Epub 2012 Feb 28. PMID: 22389682; PMCID: PMC3289640.

Methods:

Speech collection: Please clarify how many occurrences (words) for each corner vowel /a/, /i/ or /u/ were used for statistical analyses. Were all these values used for final statistical analyses or only the average value (median value) for each parameter and corner vowel?

Speech analysis: Since the calibration of microphone for loudness was not performed, the measurement if intensity cannot provide valid results. Please delete this measurement from the whole article.

Results:

Please add units for each acoustic measurement used in Table 2 or within the definition of each measurement in the speech analyses chapter.

Discussion:

Why was the younger group of healthy controls used? The using of ANOVA test here is not representative. The t-test between healthy controls and ALS without dysarthria is likely to be significant based on presented median values: ALS with dysarthria (median age value 60.0 [25th percentile 50.0; 75th percentile 66.0] years), ALS without dysarthria 62.5 [48.5;67.0], healthy controls 54.5 [42.0;60.5]. The age-dependence of many acoustic measures, including formants and vowel space area has been demonstrated in previous studies even in healthy controls. See for example:

Kent RD, Vorperian HK. Static measurements of vowel formant frequencies and bandwidths: A review. J Commun Disord. 2018 Jul-Aug;74:74-97. doi: 10.1016/j.jcomdis.2018.05.004. Epub 2018 Jun 1. PMID: 29891085; PMCID: PMC6002811.

Tykalova T, Skrabal D, Boril T, Cmejla R, Volin J, Rusz J. Effect of Ageing on Acoustic Characteristics of Voice Pitch and Formants in Czech Vowels. J Voice. 2021 Nov;35(6):931.e21-931.e33. doi: 10.1016/j.jvoice.2020.02.022. Epub 2020 Mar 31. PMID: 32245663.

Therefore, I recommend authors to recruit a better age-matched healthy control group of participants or at least to improve discussion based on results in the study published by Tykalova T et al. (2021). This study is based on a large sample of healthy controls of both genders and covers the same acoustic measurements, including f0, F1, F2, VSA and vowel duration.

6. PLOS authors have the option to publish the peer review history of their article (what does this mean?). If published, this will include your full peer review and any attached files.

Reviewer #1: No

Reviewer #2: No

---

## [Author Response · Author response to Decision Letter 0]

9 Aug 2023

We sincerely thank the anonymous reviewers, the Associate Editor, and the Editor-in-Chief for their valuable comments. We have carefully addressed the reviewers’ observations and suggestions and revised our paper accordingly. Detailed responses to the reviewers’ comments are provided as an attached file. The original reviewer’s comments are in black, and our responses are provided in blue. The appropriate changes in the annotated manuscript (PONE-D-23-09560) are highlighted in yellow.

---

## [Decision Letter · Decision Letter 1]

21 Sep 2023

Subclinical articulatory changes of vowel parameters in Korean amyotrophic lateral sclerosis patients with perceptually normal voices

PONE-D-23-09560R1

Dear Dr. Choi,

We’re pleased to inform you that your manuscript has been judged scientifically suitable for publication and will be formally accepted for publication once it meets all outstanding technical requirements.

Kind regards,

Kyung-Wan Baek, Ph.D.

Academic Editor

PLOS ONE

Additional Editor Comments (optional):

The manuscript that was reviewed has been revised to a standard that is appropriate for publication in PLoS One.

Reviewers' comments:

Reviewer's Responses to Questions

**Comments to the Author**

1. If the authors have adequately addressed your comments raised in a previous round of review and you feel that this manuscript is now acceptable for publication, you may indicate that here to bypass the “Comments to the Author” section, enter your conflict of interest statement in the “Confidential to Editor” section, and submit your "Accept" recommendation.

Reviewer #1: All comments have been addressed

Reviewer #2: All comments have been addressed

2. Is the manuscript technically sound, and do the data support the conclusions?

Reviewer #1: Yes

Reviewer #2: Yes

3. Has the statistical analysis been performed appropriately and rigorously? 

Reviewer #1: Yes

Reviewer #2: Yes

4. Have the authors made all data underlying the findings in their manuscript fully available?

Reviewer #1: Yes

Reviewer #2: Yes

5. Is the manuscript presented in an intelligible fashion and written in standard English?

Reviewer #1: Yes

Reviewer #2: Yes

6. Review Comments to the Author

Reviewer #1: Study is well written and scientifically sound, authors have fully addressed all my concerns and therefore I suggest the manuscript for publication.

Reviewer #2: The manuscript improved significantly. I have no other comments. Thank you for the revision performed.

7. PLOS authors have the option to publish the peer review history of their article (what does this mean?). If published, this will include your full peer review and any attached files.

Reviewer #1: No

Reviewer #2: No

---

## [Editor Report · Acceptance letter]

4 Oct 2023

PONE-D-23-09560R1 

Subclinical articulatory changes of vowel parameters in Korean amyotrophic lateral sclerosis patients with perceptually normal voices 

Dear Dr. Choi:

I'm pleased to inform you that your manuscript has been deemed suitable for publication in PLOS ONE. Congratulations! Your manuscript is now with our production department. 

Kind regards, 

on behalf of

Dr. Kyung-Wan Baek 

Academic Editor

PLOS ONE